# Can China's Campaign-Style Environmental Regulation Improve the Green Total Factor Productivity?

**Mingze Du [1]**[ID]**, Tongwei Zhang [2] and Dehui Wang [1],***

1   School of Mathematics and Statistics, Liaoning University, Shenyang 110036, China; dennisdu520@163.com
2   School of Economics, Liaoning University, Shenyang 110036, China; zhangtw90@163.com
*   Correspondence: wangdehui@lnu.edu.cn

**Abstract:** The central environmental inspection policy serves as a pivotal instrument for environmental regulation in China, closely intertwined with the nation's economic and social development into a greener model. Based on the urban data of China from 2004 to 2018, this paper employs a regression discontinuity design to empirically test the inherent mechanism of the central environmental inspection policy's impact on green total factor productivity, and attempts to analyze its impact on technological progress from the perspective of a bias towards technological advancement. This study found that central environmental inspections can significantly improve green total factor productivity, the mechanism behind this improvement being through the enhancement of technological progress, while having a negative impact on technical efficiency. Additionally, we found that the impact of policies on technological progress is mainly through increasing the magnitude of technological progress, rather than favoring technological progress. The results of this research provide reasonable suggestions for the Chinese government to revise their environmental inspection system.

**Keywords:** biased technological progress; central environmental protection inspector; environmental regulation; green total factor productivity; technological progress

## 1. Introduction

Environmental deterioration is a great challenge facing the whole world at present, and the key to solving this problem lies in how the government formulates and implements reasonable and effective environmental policies and means of governance. The central environmental inspection system is a unique environmental governance method, infused with Chinese characteristics and characterized by a campaign-style approach. It involves the central government's directly affiliated environmental inspection teams conducting short-term environmental inspections on pollution issues across diverse regions of China. As the largest developing country in the world, for a long time, China's economic development policy has been guided by the pursuit of high-speed growth. The realization of this economic growth rate relies on an extensive economic development model with high pollution, high energy consumption, and high emissions. This traditional development model has freed China from the trap of the poverty cycle and laid the foundation for its economic take-off, but it has also inevitably caused environmental pollution [1–4].

At present, China has the largest energy system in the world, in which fossil energy consumption occupies a dominant position, and has surpassed the United States to become the largest greenhouse gas emitter in the world [5,6]. The traditional concept of economic development and the path of long-term dependence on development has led to the subordination of environmental policies to economic growth goals, which has hindered the transformation of the existing green economic growth model and the achievement of pollution reduction targets. With the depletion of resources and the deterioration of natural ecology, it is an inevitable choice for China to seek a sustainable green economic development model under the background of global climate change. China has formulated

a series of environmental policies aimed towards shifting the goal of economic growth from high-speed growth to the pursuit of higher quality development; it also made carbon reduction commitments to achieving "Peak carbon by 2030" and "Carbon neutrality by 2060", and issued a series of related work plans and green transformation plans, providing a direction for comprehensively promoting the transition of China's economy to a sustainable model.

Current research has shown that long-term economic growth is driven by increases in factor input and factor productivity, and that total factor productivity improvement is an important measure of economic quality [7–9]. In the context of global climate change and energy and environmental constraints, the green total factor productivity derived from the inclusion of pollutant emissions and energy elements in the TFP growth-accounting framework can more accurately estimate whether this economic development approach meets the dual requirements of energy saving and emission reduction alongside economic growth [10]. In related studies, green total factor productivity levels are often used to reflect the harmony between green productivity growth, sustainable economic growth and environmental protection, and economic development [11,12]. In this paper, green total factor productivity is regarded as the core index for measuring the development of green economy.

There are differing opinions in the literature on the role of environmental regulation in promoting a green economy. Hao et al. [13] believes that proper environmental regulation can achieve a win-win situation of economic prosperity and ecological improvement. Zhuo et al. [14] believes that cross-regional environmental protection mechanisms can enhance GTFP, reduce energy intensity, and reduce carbon emissions while promoting urban innovation and industrial upgrading. Fan et al. [15] examined the role of environmental regulation in promoting green total factor productivity from a spatial correlation perspective, arguing that environmental regulation can indirectly promote total factor productivity growth by promoting green technology; however, there is spatial heterogeneity. The promotion in the eastern and central regions of China is significant, while the role of the western region is not significant. Some scholars argue that environmental regulations are not conducive to the development of a green economy. One study from Yuan and Xiang [16], based on a study of China's manufacturing sector, argues that in the short term, environmental regulations promote labor productivity and energy efficiency, among other things; but in the long run, green total factor productivity aside, environmental regulations will only improve energy efficiency and hinder productivity growth. Tian and Feng [17] studied the effects of different types of environmental regulations on GTFP and concluded that different types of environmental regulations have different internal structural effects: environmental regulation of the command and control type can promote GTFP by guiding technological innovation and optimizing the industrial structure; however, market-based environmental regulation has a negative impact on GTFP by inhibiting technological innovation, reducing the efficiency of the industrial structure and resource allocation. Li et al. [18] studied market-based environmental regulations and found that carbon trading in China promotes carbon emission reduction, but has no significant impact on industrial output; however, it has no significant effect on the improvement of GTFP.

Technological progress is key to saving energy and reducing emissions and the realization of a green and sustainable economic development model [19–21], and it is also an important way to promote the transformation of economic growth models [22–24]. The most important path to take for China's environmental protection and sustainable economic development is to adopt environmental regulation policies that can promote the progress of green technology, to achieve the dual objectives of energy saving and emission reduction and green economic transformation [25–27].

Moreover, the assumption that the substitution elasticity between labor and capital is one and that technological progress is neutral is not applicable to the present research. Hicks [28] put forward the concept of biased technological progress, and considered that technological progress is more conducive to the improvement of the marginal output of a

certain factor, resulting in the deflection of the tangent line of the isoquant curve, that is, technological progress towards the direction of this element or a bias towards this element. Kang et al. [29] recognizes that new technologies may favor energy conservation, pollution reduction, and economic output, and that promoting green technological progress is key to improving the quality of economic development. According to Liu et al. [30], the technological progress of China's manufacturing industry is biased, and environmental regulation has a significant threshold effect on technological progress. Meng et al. [31] argue that environmental regulation and environmentally friendly technological progress have important implications for reducing air pollution and green economic development in China, and that environmental regulation can strengthen technological progress in favor of environmental development. Zhou et al. [32] analyzed the impact of different types of environmental regulations on technological progress biases, and concluded that government-regulated environmental regulations promote technological progress in energy conservation and emission reduction, and market-based environmental regulations promote technological progress in energy conservation, but that informal environmental regulation can only accelerate the technical progress of environmental protection. Sun et al. [33] argue that the green output preference of technological progress is the key factor driving the green transition of mariculture in China, and that environmental regulation has a U-shaped relationship with green output preferences. Some scholars think that the effect of environmental regulation on technological progress biases is not clear. For example, Song et al. [34] think that weak environmental regulations have no significant effect on the technological progress of environmental biases.

Faced with the current environmental pollution, many countries have enacted relevant environmental protection laws to protect people's environmental rights and reverse the status quo of pollution [35,36]. The development of China's environmental governance system has formed a governance framework based on government regulatory tools, guided by market-oriented tools, and supplemented by information disclosure tools [37,38]. Among them, the central environmental inspection system, as a government regulatory environmental regulation policy, is the most official and strongest regulatory policy among all environmental regulation tools in China. Analyzing the changes in China's green total factor productivity before and after these environmental inspections can effectively promote China's economic transformation process. For green total factor productivity, technological progress is the driving force behind its growth. Therefore, finding out whether the central environmental protection inspection policy can effectively promote green technological progress and reduce biased technological progress is also the main objective of this paper.

## 2. Policy Background and Research Hypothesis

Addressing the challenge of formulating environmental policies that simultaneously promote economic development and safeguard the environment, in accordance with national circumstances, is an imminent and intricate matter in need of urgent resolution. As the world's largest developing country, China's environmental protection system has developed into a governance strategy based on government-regulated policy instruments, complemented by market- and public-participatory environmental regulations [39]. However, the government-regulated, i.e., "command and control"-oriented environmental regulation strategy is highly susceptible to problems such as competition among local governments, economic pressure, collusion between the government and enterprises, falsification of pollution data, or formalized governance [40–42]. The emergence of these problems has seriously weakened the corrective effect of environmental regulatory policies on environmental externalities.

In order to address the impact of these issues on environmental regulation, the Chinese government has issued a new environmental protection law, which has been described by many media reports as the strictest in history [43]. Moreover, in order to supervise the implementation of this new environmental protection law by local governments, the central government initiated the Central Environmental Protection Supervision scheme in

2016 to "supervise" the effectiveness of local governments' environmental governance in a "campaign-style" manner; and, in 2019, the central government promulgated provisions amongst the work of the Central Ecological Environmental Protection Inspectorate, which explicitly stated in the form of internal regulations of the Communist Party of China (CPC) that environmental protection inspections would be carried out on a regular basis once every five years. Since then, China's environmental regulatory system has entered a phase of dual constraints between government-controlled policies and environmental protection inspections.

Unlike traditional environmental regulatory measures, the central environmental inspection policy takes into account various levels and aspects of environmental regulation, which not only has the authority of the state but also ensures the political and social interaction of public participation. The central environmental protection inspection policy is no longer based on a hierarchical government, but rather on empowering environmental protection regulatory agencies with higher supervision rights and the authority represented by the central government, horizontally integrating environmental protection regulatory agencies, and constructing a more targeted supervision system. Through "party and government shared responsibility" and public supervision, the centralized governance model is regularly under the direct management of the central government [44,45]. From the preparation of inspectors, the presence of inspection teams, and the reporting of inspection sites to the feedback of inspection teams, the handing over of inspection issues, and the rectification of inspection issues, the process of the central environmental protection inspectors (inspecting each province under inspection before and after a total of about one month) includes listening to the environmental work reports of the inspected provinces, consulting the information on the local environmental work, visiting and inquiring, accepting reports, and environmental spot checks. After the completion of this inspection, the province under supervision shall submit their environmental rectification plan to the central committee and State Council within 30 days and complete the rectification requirements within half a year. At the same time, the contents of their rectification plan shall be disclosed to the public. In addition, the Central Ministry of Ecology and Environmental Protection has set up special channels for environmental reporting, including telephone and email accounts, to expand the "government–society communication" model for environmental issues; this allows the central government to co-ordinate the planning and inspection of environmental protection at all levels of local governments, to a certain extent avoiding the problem of information asymmetry, and reducing the possibility of local governments hiding or falsely reporting environmental problems. In addition, the central environmental inspection scheme and inspection team has adopted a "looking back" approach to ensure that this innovative environmental regulation system adheres to its original problem-oriented design. "Looking back" means that the central environmental inspection team will focus on monitoring the effectiveness of the plans for the rectification of environmental inspections at all levels of government, which have been vetted by the central committee and the State Council, to supervise the environmental protection results of the provinces inspected [46]. This kind of "looking back" system designed to elevate environmental protection into becoming a political mission has enhanced the authority and public participation of environmental inspectors; it also promotes political–social interactions and the sustainable governance of environmental regulation, and promotes the effectiveness of environmental regulation.

The implementation of central environmental supervision represents a strategic adjustment of environmental policies from the perspective of the Chinese government's environmental regulation, from the traditional "supervision of enterprises" to the present "supervision of enterprises" and "supervision of government". However, as a newly implemented environmental regulation policy, this policy's ability to balance environmental governance and green total factor productivity is an important theoretical underpinning for China's strategy of normalizing the implementation of central environmental inspections.

At present, existing studies on central environmental inspection are mainly from the perspective of the policy's effect on corporate environmental governance and air pollution

control, and whether it can promote the development of green economy. Wang et al. [47] discovered through their analysis of enterprise-level pollution data in China from 2011 to 2018 that since the implementation of the central environmental inspection policy, the number of polluting enterprises has decreased by 48%, and emissions have significantly reduced. Research by Feng et.al [48] shows that environmental inspections have greatly improved air quality, but that their subsequent impact is waning. Similarly, Zhao Zhang and Wang's [49] research shows that such campaign-style treatment is effective in reducing pollution, but that its impact on pollutant emissions is only temporary. On the contrary, Jia and Chen [50] believe that the central environmental inspection system can play a positive role in improving environmental performance, and that this role of improvement does not disappear after the inspection. According to Cheng and Yu [51], the central environmental watchdog scheme has greatly promoted green technology innovation in pollution-intensive industries and improved fossil energy efficiency, but rather than driving green innovation to reduce pollutants, it is crowding out other technological innovations. Kopyrina, Wu, and Ying's [52] study demonstrated that this policy can have has a long-term positive effect on corporate green innovation in the form of green patents. He and Geng [53] argue that not all areas inspected have seen a significant reduction in their air quality index since the implementation of the central environmental inspection policy, and that individual areas and individual pollutants have not. Pan, Yu, Hong, and Chen [43] believe that the central environmental inspection policy can significantly improve green economic growth and that this effect is sustainable.

Based on the analysis of the aforementioned policy background, this article proposes the following hypotheses.

① The impact of the central environmental inspection policy on green total factor productivity:

The central environmental inspection policy can provide strong supervision of enterprises and industries to change their environmental decision-making and incorporate environmental management into their corresponding production and operation processes. During the implementation of this policy, production and business operators in the region received policy notices, changed their environmental strategies, increased their environmental investment quotas, actively developed and introduced green production technologies, and proactively turned to green production. The post-inspection responsibility system adopted by the central environmental inspection policy can hold government officials accountable for their dereliction of duty in supervising local industry production and environmental governance. This system has stimulated local officials to actively respond to their region's economic green transformation and improved the implementation effect of this environmental protection system. Therefore, this article proposes the following hypothesis:

**Hypothesis 1.** *The central environmental protection inspection policy can significantly improve the productivity of green total factor production.*

② The impact of the central environmental inspection policy on the progress of green technology:

The central environmental inspection system, with the strong authoritative attribute of the central government, endows the environmental inspection agencies with authority and effectiveness, directly promoting the strong implementation of environmental regulation. Therefore, it is recognized as a "campaign-style governance model". Through the official responsibility system, the long-standing phenomenon of "collusion between the government and enterprises" in local areas is resolved, promoting polluting enterprises to break away from their original production model, enabling the corresponding industries to upgrade their model, and increasing the introduction and adoption of environmental protection technologies in production processes. At the same time, for enterprises in the inspected areas with green research and development attributes, their financing and research and development investment will receive more attention and support, accelerating the

commercialization of corresponding green technologies. By avoiding the "formalization" of environmental problems, the sustainable development process of the regional economy is promoted. Therefore, this article proposes the following hypothesis: by avoiding the "formalization" of environmental problems, the sustainable development process of the regional economy is promoted.

**Hypothesis 2.** *The central environmental protection inspection policy can significantly improve the level of green technological progress.*

③ The impact of central environmental inspections on biased technological advancements:

Compared to neutral technological progress, biased technological progress refers to the marginal output improvement of a specific production factor. Under the guidance of environmental regulations, technological progress is no longer simply a matter of improving technical levels, but has a certain bias. During the process of technological research and development, companies will control the pollution generated by their technology, minimize environmental costs, and gradually shift technological progress towards green development. However, unlike other environmental regulatory policies, the central environmental protection inspection system, as a new type of environmental policy with Chinese characteristics, has the short-term governance characteristics of "authoritative attributes" and "campaign-style governance". This forces companies to deal with pollution output from the production end, increase their investment in pollution control, and change their current pollution situation. However, fundamentally reversing the biased technological progress of corporate production requires improving the production nature of companies from the source, adopting green production models, introducing green production equipment, and making production factors more inclined towards economic output while reducing pollution output. Therefore, guiding biased technological progress is difficult to implement in the short term and requires fundamentally transforming the production mode of companies from "polluting" to "green" production. Therefore, this article proposes the following hypothesis:

**Hypothesis 3.** *The central environmental protection inspection policy cannot significantly improve the level of biased technological progress.*

The aforementioned findings indicate the existence of certain controversies surrounding the effectiveness of existing research on the implementation of the central environmental inspection, particularly regarding the impact on green total factor productivity. Additionally, there appears to be a scarcity of studies focusing on the decomposition of the impact pathway of green total factor productivity. The implementation of central environmental inspection will exert pressure on local governments and polluting enterprises in the provinces being inspected, and will be sudden and random. Most of the cases accepted by the central environmental inspection group come from public reports and other means. The inspection projects also have a certain suddenness and confidentiality. Local governments do not make "response" plans for the inspection projects in advance, therefore, the central environmental inspection conforms to the idea of a quasi-natural experiment. This paper uses a regression discontinuity design to analyze the impact of the innovative environmental regulation policy of central environmental protection inspections on green total factor productivity, and decompose the green total factor productivity index into technical efficiency and technological progress, and will continue to deconstruct the technical progress index into a scale of technical progress and biased technical progress. Our aim is to analyze in detail the impact of "campaign-style" environmental regulations on green total factor productivity and provide recommendations for China's transition towards a green economy and achieving pollution reduction targets.

### 3. Methods and Data

#### 3.1. The Measurement and Decomposition of the Green Total Factor Productivity

Stochastic frontier approaches (SFAs) and data envelopment analyses (DEAs) are the main methods used to measure total factor productivity in academic circles; this can avoid the structural deviation caused by the misunderstanding of production function, such as occurs in the traditional accounting method and SFA method [54–56]. In view of this, the traditional DEA model cannot solve the problem of relaxation variable, and when the efficiency of efficient decision-making units is 1, the units cannot be distinguished well. Therefore, this paper adopts the Tone and Tsutsui (2010) SBM model, which considers the relationship between the input, output, and adverse pollution output, and which can solve the problem of slack in efficiency evaluation.

In this study, we construct a non-radial and non-oriented relaxation-based directional distance function [57], combining the work of Fare [58] to calculate the Malmquist index, and then decomposing the Malmquist index according to Fare [59]. It is decomposed into changes towards green technology and green efficiency, and further decomposed into the magnitude of technological changes and the biased technological change index. In this study, we regard the Chinese city as a production decision unit (DMU) to construct the optimal production technology boundary. Assuming that each city uses $N$ inputs to obtain $M$ expected outputs and $U$ unexpected outputs, the production process can be expressed as:

$$D(x^{t,k}, y^{t,k}, b, t, kg^x, g^y, g^b) = Max \frac{\frac{1}{N}\sum_{n=1}^{M}\frac{s_n^x}{g_n^x} + \frac{1}{M+U}(\sum_{m}^{M}\frac{s_m^y}{g_m^y} + \sum_{u=1}^{U}\frac{s_u^b}{g_u^b})}{2}$$

$$s.t. \begin{cases} \sum_{t=1}^{T}\sum_{k=1}^{K}\lambda_k^t x_{kn}^t + s_n^x = x_{k'n}^t, \forall n \\ \sum_{t=1}^{T}\sum_{k=1}^{K}\lambda_k^t y_{km}^t - s_m^y = y_{k'm}^t, \forall m \\ \sum_{t=1}^{T}\sum_{k=1}^{K}\lambda_k^t u_{kn}^t + s_u^b = b_{k'u}^t, \forall u \\ \sum_{k=1}^{K}\lambda_k^t = 1, \lambda_k^t \geq 0, \forall k \\ s_n^x \geq 0, \forall n; s_m^y \geq 0, \forall m; s_u^b \geq 0, \forall u \end{cases} \quad (1)$$

where $x^{t,k}$, $y^{t,k}$, and $b^{t,k}$ respectively represent the factor input, economic output, and unexpected output of $k$-city in $t$-period; $g^x$, $g^y$, and $g^b$ indicate the direction vector of the input of production factors, the increase in economic output, and the decrease in undesired output, respectively; and $s_n^x$, $s_m^y$, and $s_u^b$ respectively represent the relaxation vectors for the input of production factors, the economic output, and the unexpected output.

Combining the SBM non-radial model with undesirable outputs, a distance function of the Malmquist index is constructed. Based on the Malmquist index decomposition method, the *GTFP* growth rate is decomposed into technological changes and changes in efficiency. Furthermore, the technological changes are further decomposed into three parts: output-biased technological changes (OBTCs), input-biased technological changes (IBTCs), and the magnitude of the technological change (MATC).

$$GTFP_k^{t,k+1} = \sqrt{\frac{D^t(x^{t+1}, y^{t+1}, b^{t+1}; g)}{D^t(x^t, y^t, b^t; g)} \times \frac{D^{t+1}(x^{t+1}, y^{t+1}, b^{t+1}; g)}{D^{t+1}(x^t, y^t, b^t; g)}} = \sqrt{N^{t+1} \times N^t} \quad (2)$$

In Formula (2), $N^{t+1}$ represents the change in green technology efficiency from period $t$ to period $t+1$ under the technological conditions at period $t$, and $N^t$ also represents the change in technology efficiency from period $t$ to period $t+1$ under the technological conditions at period $t+1$. $GTFP_k^{t,t+1}$ is the geometric mean of $N^{t+1}$ and $N^t$. When $GTFP_k^{t,t+1} > 1$, it means that the *GTFP* increases from $t$ period to $t+1$ period. When

$GTFP_k^{t,t+1} < 1$, it means that the *GTFP* decreases from $t$ period to $t+1$ period. According to the Malmquist index decomposition method, *GTFP* growth is decomposed into technical changes and efficiency changes as follows:

$$GTFP_k^{t,t+1} = TC_k^{t,t+1} \times EC_k^{t,t+1}$$
$$= \sqrt{\frac{D^t(x^{t+1}, y^{t+1}, b^{t+1}; g)}{D^t(x^t, y^t, b^t; g)} \times \frac{D^{t+1}(x^{t+1}, y^{t+1}, b^{t+1}; g)}{D^{t+1}(x^t, y^t, b^t; g)}} \times \frac{D^{t+1}(x^{t+1}, y^{t+1}, b^{t+1}; g)}{D^t(x^t, y^t, b^t; g)} \tag{3}$$

where $TC_k^{t,t+1}$ represents the technological change of the *k*th DMU during the period from $t$ to $t+1$, that is, the movement of the technological frontier; and $EC_k^{t,t+1}$ represents the change in relative efficiency.

After decomposing *MI*, *TC* is decomposed into the magnitude of technological change (*MATC*) and biased technology change (*BTC*) according to the decomposition method of Fare, Grosskopf, and Margaritis (2006). Further, the technology bias index can be decomposed into input-biased technology change (*IBTC*) and output-biased technology change (*OBTC*). This specific decomposition process is as follows:

$$TC_k^{t,t+1} = MATC_k^{t,t+1} \times BTC_k^{t,t+1}$$
$$= \frac{D^t(x^{t+1}, y^{t+1}, b^{t+1}; g)}{D^{t+1}(x^{t+1}, y^{t+1}, b^{t+1}; g)} \times \sqrt{\frac{D^t(x^t, y^t, b^t; g)}{D^{t+1}(x^t, y^t, b^t; g)} \times \frac{D^{t+1}(x^{t+1}, y^{t+1}, b^{t+1}; g)}{D^t(x^{t+1}, y^{t+1}, b^{t+1}; g)}} \tag{4}$$

Furthermore, the biased technology index is decomposed as follows:

$$BTC_k^{t,t+1} = IBTC_k^{t,t+1} \times OBTC_k^{t,t+1}$$
$$= \sqrt{\frac{D^t(x^t, y^t, b^t; g)}{D^{t+1}(x^t, y^t, b^t; g)} \times \frac{D^{t+1}(x^{t+1}, y^{t+1}, b^{t+1}; g)}{D^t(x^{t+1}, y^{t+1}, b^{t+1}; g)}}$$
$$= \sqrt{\frac{D^{t+1}(x^t, y^t, b^t; g)}{D^t(x^t, y^t, b^t; g)} \times \frac{D^t(x^{t+1}, y^t, b^t; g)}{D^{t+1}(x^{t+1}, y^t, b^t; g)}} \times \sqrt{\frac{D^t(x^{t+1}, y^{t+1}, b^{t+1}; g)}{D^{t+1}(x^{t+1}, y^{t+1}, b^{t+1}; g)} \times \frac{D^{t+1}(x^{t+1}, y^t, b^t; g)}{D^t(x^{t+1}, y^t, b^t; g)}} \tag{5}$$

That is,

$$TC_k^{t,t+1} = MATC_k^{t,t+1} \times IBTC_k^{t,t+1} \times OBTC_k^{t,t+1} \tag{6}$$

where *MATC* represents the magnitude of technological change, which is the neutral transfer of the technological frontier, and *BTC* represents the bias of technological change, which is the "non-neutral" transfer of the technological frontier. *IBTC* and *OBTC* reflect the impact of input and output changes on technological progress. If *IBTC (OBTC)* > 1 (<1), it indicates that there is progress in input-biased technology (retrogression). When *IBTC* and *OBTC* = 1, it indicates that the technical change is Hicks neutral.

*3.2. Establishment of the Impact Model of the Central Environmental Protection Inspection Policy on Green Total Factor Productivity*

A regression discontinuity design (RDD) is a quasi-natural experiment-based measurement method applied to policy assessments. Compared with the instrumental variable method and propensity score-matching method, a regression discontinuity design is closer in nature to randomized trials and thus has become the preferred method in current causal identification studies [60,61].

The regression discontinuity design method is now widely used in labor economics, health economics, political economy economics, and environmental economics. According to the regression discontinuity design concept, if variables such as the green total factor productivity and their decomposition items suddenly change before and after the central environmental inspection policy, and other important variables that affect the green total factor productivity and their decomposition are not suddenly changed, then we have reason to believe that the sudden change in variables such as the green total factor productivity is caused by the central environmental inspection, i.e., that it has an effect on the green total factor productivity indices and their decomposition items.

According to the regression discontinuity design principle, this paper establishes the following model for calculating the impact of the central environmental protection inspection policy on green total factor productivity:

$$LnGTFP_{it} = \alpha_0 + \alpha_1 CEPI_{it} + \alpha_2 f(x) + \alpha_3 CEPI_{it} f(x) + \lambda X_{it} + \delta_i + \mu_t + \varepsilon_{it} \qquad (7)$$

where $i$ represents the central environmental protection inspection city and $t$ represents the year; $GTFP_{it}$ is the green total factor productivity of the inspected city in the current year, which is logarithmized in this section to reduce heteroscedasticity and ensure stability; $CEPI_{it}$ is a dummy variable representing the central environmental protection inspector. $CEPI$ is 0 for $i$ city in the year before the central environmental protection inspector's inspection, and 1 for $i$ city in the year after the central environmental protection inspector's inspection. The first pilot of China's central environmental protection inspection policy and the implementation of the first round of inspections all started after 2016, so the time of the exogenous impact proxy variable of regression discontinuity design model was set to 2016. $f(x)$ is a polynomial function with $x$ as the independent variable; $X_{it}$ is a series of control variables; $\delta_i$ is the regional fixed effect; $\mu_t$ is the time-fixed effect; and $\varepsilon_{it}$ is the random disturbance term. In this formula, the coefficient $\alpha_1$ is the difference in green total factor productivity before and after the central environmental protection inspection.

In order to explore the impact mechanism of central environmental protection inspections on the green total factor productivity in detail, we decomposed $GTFP$ into technical efficiency $EC$ and green technological progress $TC$, and used the model idea of formula to establish the following model, so as to analyze the mechanism of the impact of the central environmental protection inspections on these two decomposition items.

$$LnEC_{it} = \beta_0 + \beta_1 CEPI_{it} + \beta_2 f(x) + \beta_3 CEPI_{it} f(x) + \lambda X_{it} + \delta_i + \mu_t + \varepsilon_{it} \qquad (8)$$

$$LnTC_{it} = \gamma_0 + \gamma_1 CEPI_{it} + \gamma_2 f(x) + \gamma_3 CEPI_{it} f(x) + \lambda X_{it} + \delta_i + \mu_t + \varepsilon_{it} \qquad (9)$$

In order to explore whether central environmental protection inspections can change the direction of technological progress, this paper further decomposes the green technological progress $TC$ into the output-biased technological change $OBTC$, the input-biased technological change $IBTC$, and the magnitude of technological change $MATC$. If the central environmental protection inspection cannot significantly change the technological progress bias, this indicates that the environmental regulation has not fundamentally promoted the green transformation but rather the "short-term effect" of environmental governance.

### 3.3. Variables and Data

Since the sample selected in this paper is a sample of Chinese cities, based on the availability and integrity of data, we selected panel data from a total of 267 cities from 2004 to 2018. The relevant data of the variables explained above and the control variables are from the *China Urban Statistical Yearbook*, *China Statistical Yearbook*, and the *Province Statistical Yearbook*. Any missing values in the process of data collation were filled using the mean method.

(1) Explanatory variables: the explanatory variables are the green total factor productivity $GTFP$ and the technical efficiency $EC$, the green technological progress $TC$, the biased technological progress after the decomposition of green technological progress $OBTC$ and $IBTC$, and the magnitude of technological change $MATC$ of 267 cities in China, measured by using the non-radial super-efficiency SBM model considering the undesirable output. The specific labor input, capital input, energy input, desirable output, and undesirable output indicators involved are shown in Table 1.

**Table 1.** Main input and output variables and methods of calculating the GTFP.

| Indicator Category | Indicator Name | Index Content |
| --- | --- | --- |
| Input indicators | Capital input | The capital stock of fixed assets in each city, calculated via the perpetual inventory method |
| | Labor input | The number of people employed at the end of the year in each city |
| | Energy input | Total annual electricity consumption |
| Desirable output | Economic output | Gross domestic production at constant 2003 prices |
| Undesirable output | Environmental pollution index | The concentration of industrial dust, industrial wastewater, industrial sulfur dioxide, and Urban PM 2.5, calculated via the entropy weight method |

The specific input–output indicators are selected as follows:

(2) Explanatory variables: the core explanatory variable is the central environmental protection inspection $CEPI$, and the remaining control variables are foreign direct investment $FDI$, human capital $HC$, industrial structure $IS$, fiscal revenue $FR$, and R&D capacity $IC$. The details of the variables are as follows:

① Foreign direct investment (FDI): there are two opposite hypotheses of a "pollution halo" and "pollution refuge" in foreign investment. When foreign investment brings advanced technological production, it will also transfer pollution to the local area. Therefore, the proportion of foreign direct investment in the local GDP of each city is selected to represent this indicator.

② Human capital (HC): the higher the level of human capital, the higher the level of industrial labor quality and knowledge. This indicator is represented by the number of college students per million population.

③ Industrial structure (IS): if a city takes a polluting industry as its pillar industry, the more dependent it is, the more difficult it is to transform and upgrade the polluting industry. This indicator is represented by the proportion of the secondary industry in the local GDP.

④ Fiscal revenue (FR): the general fiscal revenue of each city in the current year is used to represent the local tax revenue and other economic conditions.

⑤ R&D capability (IC): the support of cities for scientific undertakings can effectively ensure their R&D capability and application of green technologies and their introduction and adoption of advanced production technologies. The R&D capability is characterized by the proportion of local fiscal expenditure on scientific undertakings in the local GDP.

## 4. Empirical Analysis

### 4.1. Analysis of the Mechanism of the Central Environmental Inspection Policy's Impact on Green Total Factor Productivity

Before employing the regression discontinuity design, this article examines the relationship between the central environmental inspections and the GTFP to ensure that there are abrupt changes in the GTFP of the cities around the year of their inspections. Figures 1 and 2 show the impact of the Central Ecological Environmental Protection Inspectorate on the change in efficiency (EC) and technological change (TC) of the green total factor productivity and its decomposition items (magnitude of technological change, MATC; input-biased technological change, IBTC; and output-biased technological change, OBTC) in the year of the inspection, respectively. On the left of the dotted line is the year

before the implementation of the central environmental inspection, and the dotted line is the year 2016. This paper mainly analyzes the "jump" of the variables before and after the implementation of the environmental protection inspection policy, and the trend in the following years.

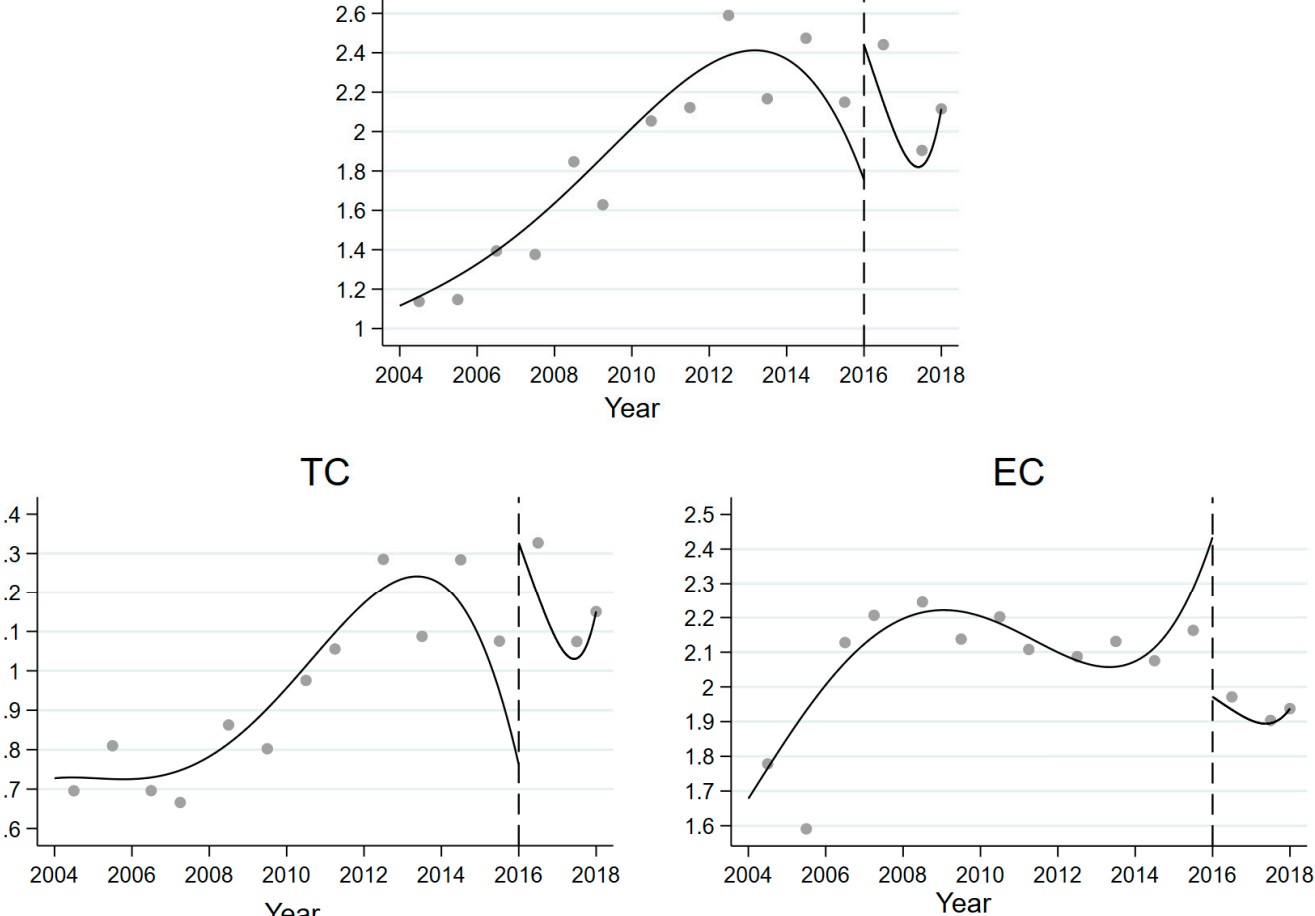

**Figure 1.** Regression discontinuity design analysis of the GTFP index's decomposition items in relation to the central environmental inspection policy.

As can be seen from Figures 1 and 2, there is a significant jump in the GTFP value in Chinese cities before and after the central environmental inspection; this shows that the implementation of central environmental inspections has a significant effect on the promotion of GTFP. In terms of the decomposition items of GTFP, the central environmental inspection policy had a marked effect on the improvement of the TC and a dampening effect on the EC, resulting in a drop in EC in the short term.

From the perspective of the decomposition items of TC, there is a positive effect of the Central Ecological Environmental Protection Inspectorate on the MATC and OBTC. There is no obvious jump in the year of the cut-off point for the IBTC, but there is an obvious upward trend after the cut-off year; this shows that the central environmental inspection policy has a positive effect on the decomposition items of TC. From the above analysis, we know that TC is the core driving force of GTFP promotion, so we can draw a rough conclusion from the diagram that the Central Ecological Environmental Protection Inspectorate can effectively promote GTFP growth through TC. However, due to the systemic design of the Central Ecological Environmental Protection Inspectorate, when the inspected areas are faced with environmental inspections, this will exert a certain pressure on polluting enterprises and local authorities. Judging from the actual situation of the central environmental

protection inspection team, the handling of environmental pollution incidents after the inspection team's arrival not only includes the environmental protection departments of local governments, but also involves the joined forces of the local development department, Reform Commission, the Public Security Bureau, the Urban Construction Bureau, the Urban Management Bureau, and many other departments to carry out joint law enforcement and environmental supervision. Pollution-related enterprises found by inspection teams and reported by the masses will face the consequences of production suspension, rectification, technological upgrading, and transformation, as well as environmental fines, regarding their production processes, the efficiency of the coordination between government and enterprises, the efficiency of their production management, and other aspects of the impact of a decrease in technical efficiency. The reprimanding of the companies involved after environmental inspections and the over-adaptation of new technologies also led to a short-term decline in indicators such as GTFP and TC after increasing the environmental inspection cut-off point. As Figures 1 and 2 only show the initial identification of the impact of the central environmental inspection policy on the GTFP, and the graphic is a rough analysis based on the average of the samples, the next step is to conduct a regression discontinuity design analysis to explore the GTFP-enhancing effect of the Central Ecological Environmental Protection Inspectorate.

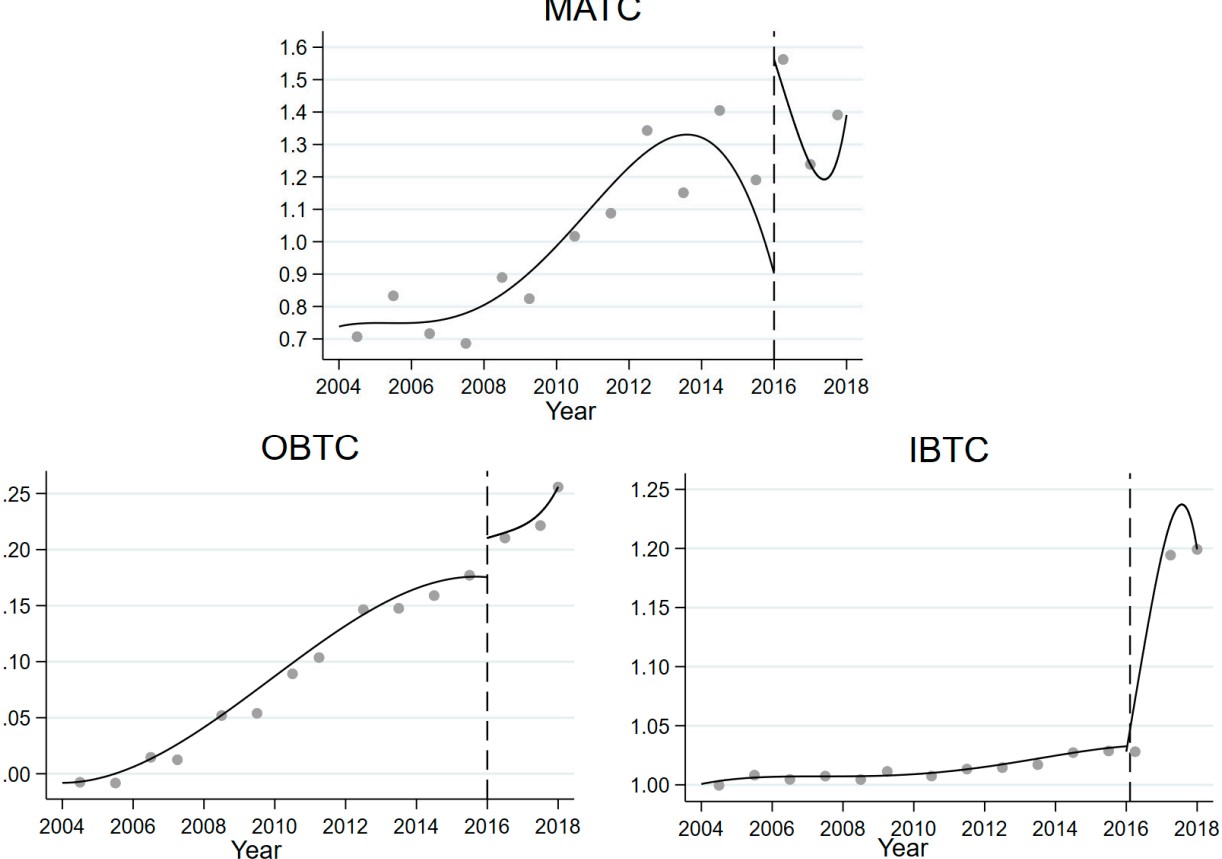

**Figure 2.** Regression discontinuity design analysis of the TC index's decomposition items in relation to the central environmental inspection policy.

### 4.2. The Central Environmental Inspector's Regression Discontinuity Design Analysis of the GTFP

Table 2 shows the results of the regression discontinuity design analysis under the optimal bandwidth selection of the GTFP from the central environmental inspector. When the control variables were not taken into account, the central environmental inspection had a positive effect on the GTFP at the 10% significance level, with a coefficient of 0.063, that is,

when a city is faced the central environmental inspection, it will increase its GTFP rate by 6.313%. When variables such as the foreign direct investment, human capital, industrial structure, fiscal revenue, and R&D capacity are taken into account, the coefficient of the central environmental inspection has a positive effect on the GTFP at the 1% significance level, to 0.06361, representing an increase of 6.361% in the GTFP. Thus, whether including the control variables or simply considering the central environmental inspection, the coefficient of the central environmental inspection is significantly positive; this shows that the Central Ecological Environmental Protection Inspectorate can significantly improve the GTFP of the areas under inspection in the short term.

**Table 2.** The impact of the Central Ecological Environmental Protection Inspectorate on the GTFP.

|  | **LnTFP** | **LnTFP** |
| --- | --- | --- |
| CEPI | 0.06313 * | 0.06361 *** |
|  | (1.69) | (2.31) |
| Control variables | No | Yes |
| Time-fixed effect | Yes | Yes |
| Regional-fixed effects | Yes | Yes |
| N | 267 | 267 |

Note: * and *** indicate $p < 0.1$, and $p < 0.01$, respectively. T values are in parentheses.

In order to analyze the impact of the central environmental inspection on the GTFP in detail, this paper decomposes the GTFP index into EC and TC. As the TC was significantly positive at a level of 1%, and the TC was the main driving force of the GTFP, we further decomposed the TC into MATC, OBTC, and IBTC, with the goal of providing a detailed analysis of the TC enhancement pathways of the central environmental inspections. The breakdown of these regression results is shown in Table 3.

**Table 3.** Effects of the central environmental inspection policy on GTFP decomposition.

|  | **LnEC** | **LnTC** | **LnMATC** | **LnOBTC** | **LnIBTC** |
| --- | --- | --- | --- | --- | --- |
| CEPI | −0.101 ** | 0.109 *** | 0.097 ** | 0.005 | 0.011 |
|  | (−2.1) | (4.26) | (2.38) | (0.17) | (0.70) |
| Control variables | Yes | Yes | Yes | Yes | Yes |
| Time-fixed effect | Yes | Yes | Yes | Yes | Yes |
| Regional-fixed effects | Yes | Yes | Yes | Yes | Yes |
| N | 267 | 267 | 267 | 267 | 267 |

Note: **, and *** indicate $p < 0.05$, and $p < 0.01$, respectively. T values are in parentheses.

Table 3 shows that the central environmental protection inspection has a negative effect on EC at a 5% significance level, with a coefficient of −0.101. This may be due to the fact that when the inspected areas are faced with the entry of the central environmental inspection team, their polluting enterprises will be ordered to shut down and be rectified and fined. In serious cases, this will lead to the criminal liability of the relevant responsible persons, the production efficiency and management efficiency of the production enterprises concerned will decrease in a short period of time, and the equipment capacity of the pollution attribute in the production process will be upgraded, causing a significant negative impact on the technical efficiency of the central environmental inspection. The central environmental inspection policy has a positive impact on TC at a 1% significance level, with a coefficient of 0.109. This may be because the implementation of the central environmental inspection makes local governments pay more attention to the R&D activities of their technology; it also urges the companies involved in pollution control to improve their use and development of green technology equipment, which is further demonstrated by the results of the regression discontinuity design analysis of the TC decomposition.

For the TC decomposition items, the central environmental inspection had a significant positive effect on the MATC at the 5% level, with a coefficient of 0.097. The central

environmental inspection had positive effects on both the OBTC and IBTC, but failed the significance test. This result shows that the impact pathway of the central environmental inspection to the TC is mostly through the promotion of the magnitude of technological change. Although it has a positive effect on the biased technological change, it is not significant; that is, after the central environmental inspection, local governments promoted the development and application of green technologies, and urged polluting enterprises such as industrial enterprises to upgrade and use more environmentally friendly production and sewage equipment, which led to an increase in the magnitude of their technological changes. But the result of the biased technological change also means that the central environmental inspection policy does not have a significant impact on the input of factors of production, the desirable output and the pollution output in the production process. It shows that the improvement of the current central environmental inspection is more about the "treatment effect" of the pollution output, as the matching of production factors with inputs from the source, the technological upgrading, and the proportion of output factors have little influence on the production process.

The central environmental inspection policy is logically a type of "campaign-style" policy [50] that was initially set up to correct local governments' lack of implementation of environmental regulations and reduce the practice of incomplete enforcement, as well as for the supervision of local governments that "value the economy more than environmental protection" and the punishment of typical polluting enterprises.

The central environmental inspection policy process has a short-term authoritative quality. It is also because of the "top-down" nature of the central environmental inspection policy that administrative pressure on local party and government officials can prompt departments to take a proactive approach to joint law enforcement. Therefore, it has a good effect on the aspects of the green discharge treatment of local enterprises, the importance of the green output of these enterprises, and the pursuit of the environmental performance of the environmental protection officials. But it is also because of the "campaign-style" nature of this environmental governance that improvements are evident in the short term; however, its long-term impact may be weakened after the departure of the environmental inspection team. Therefore, to realize the "green" transformation of enterprises, it has a positive effect on the input ratio of production factors and the behavior of radically reducing relative pollution output, but this influence behavior was not significant during the study period.

*4.3. Robust Test*

This section of the study will discuss testing the sensitivity of the central environmental inspection policy to the GTFP results in terms of the continuity of the control variables and the significance of the main explanatory variables across different bandwidths.

(1)    Continuity test for GTFP control variables:

If the GTFP's main control variables also show a significant "jump" in the central environmental inspection policy year of 2016, the GTFP jump cannot then be attributed entirely to the implementation of the central environmental inspection policy, but rather to other policies or influences. Figure 3 illustrates the central environmental inspection policy continuity analysis of the main control variables for the urban GTFP in our country. A 95% confidence interval for the trend of the variables has been added to increase the likelihood of jumps for these variables in the policy implementation year. As can be seen from this figure, only the R&D level before and after the policy implementation year shows a significant upward "jump" situation; the main control variables, such as foreign direct investment, human capital level, industrial structure, and fiscal revenue, did not show significant jumps before and after the implementation of environmental supervision. The central environmental inspection policy is the main reason why the GTFP values exhibited a significant jump after the policy's implementation in 2016, namely the robustness of the regression discontinuity design results. Since graphic analyses are only rough estimates of the main control variables and cannot accurately identify the significance of changes in

the control variables, the continuity of the control variables was tested using a regression discontinuity design analysis, and these test results are shown in Table 4.

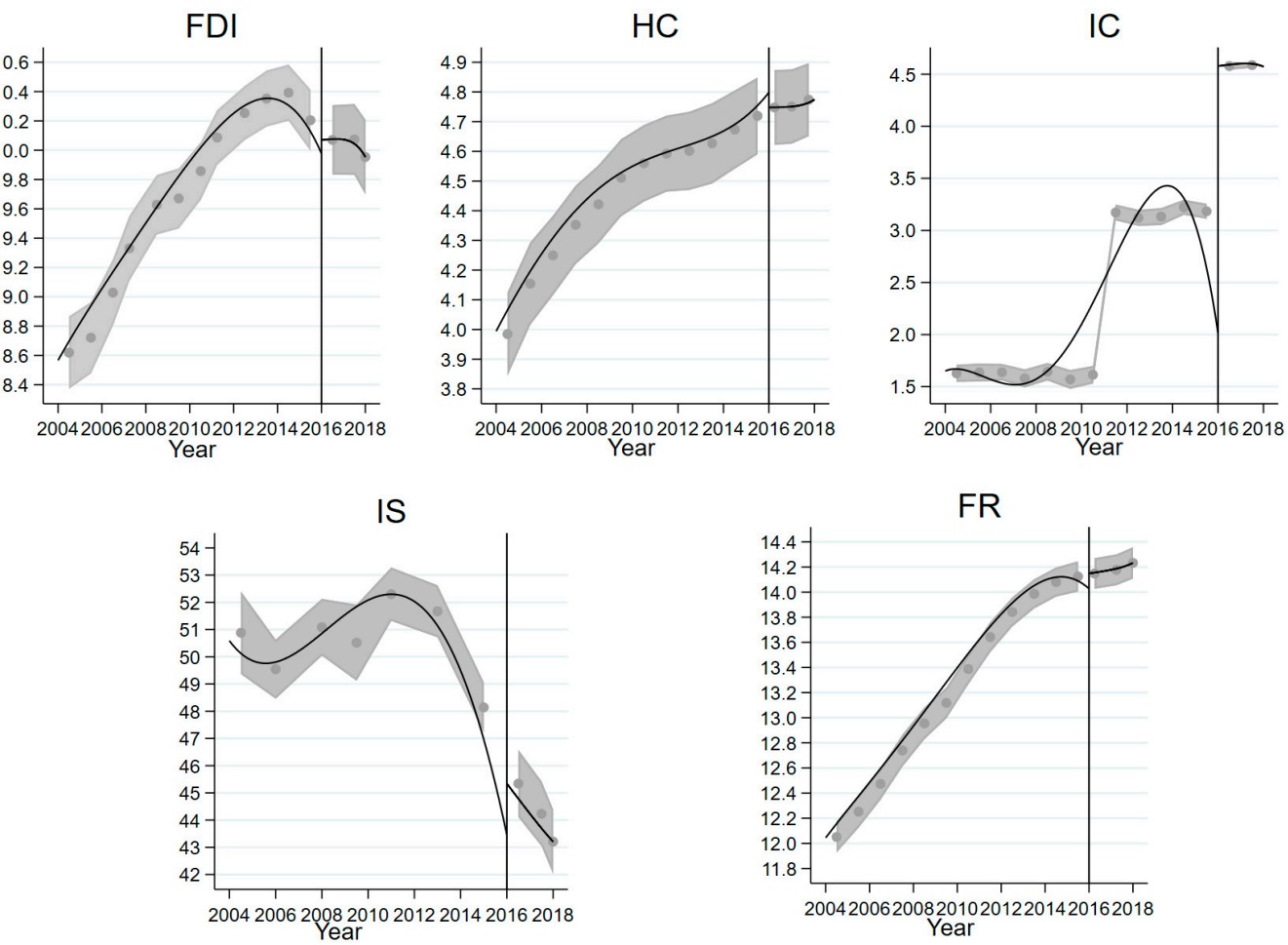

**Figure 3.** Continuity analysis of the cities' GTFP control variables.

**Table 4.** Testing for controlled variables.

|  | FDI | HC | IC | IS | FR |
|---|---|---|---|---|---|
| CEPI | −0.087 | −0.017 | 1.368 *** | 0.17 | −0.063 |
|  | (−0.47) | (−0.16) | (24.71) | (0.15) | (−0.61) |
| Control variables | Yes | Yes | Yes | Yes | Yes |
| Time-fixed effect | Yes | Yes | Yes | Yes | Yes |
| Regional-fixed effects | Yes | Yes | Yes | Yes | Yes |
| N | 267 | 267 | 267 | 267 | 267 |

Note: *** indicate $p < 0.01$. T values are in parentheses.

The results of the continuity test for the main control variables in Table 4 are consistent with the analysis in Figure 3. Except R&D, foreign direct investment, and other key control variables failed the significance test before and after the year of implementation of the central environmental inspection policy, it shows that these variables have no "jump" around the year of implementation of the policy. It further shows the robustness of the central environmental inspection policy to the regression results of GTFP and its decomposition. The change in R&D level before and after the implementation of the policy may be due to the increased investment in cleaner production technologies and the use of cleaner production equipment by the local governments inspected after the central environmental inspection policy, leading to an increase in the level of research and development. The level of R&D is

the main factor affecting the TC, while the central environmental inspection policy has a significant upward TC, that is, there is a significant short-term positive impact. So central environmental inspection policy and local government support for R&D could be a source of TC. From the previous regression results, it was found that among the decomposition terms of TC, the central environmental inspection policy had the most significant effect on MATC, it shows that the central environmental inspection policy has mainly promoted the application and R&D of equipment such as sewage treatment in the short term, but has no significant positive effect on the results of biased technological change, it also shows that the central environmental inspection policy has failed to significantly promote the "green" transformation of enterprises.

(2)    Sensitivity test of different bandwidths of green total factor productivity:

The results of the regression discontinuity design model are heavily influenced by the choice of bandwidth. The results of the regression discontinuity design analysis are estimated under the optimal bandwidth selection. Therefore, to test the robustness of the estimation results, this section provides an estimate of the central environmental inspection policy's impact on GTFP at different bandwidth levels. Table 5 shows the sensitivity test results of the coefficients under different bandwidth estimates. The results show that for different bandwidths, the central environmental inspection policy has a positive effect on the GTFP at a certain level of significance. These results are in good agreement with the above results, which shows that the conclusions of this paper are robust under different bandwidths.

**Table 5.** Estimation of coefficient variation under different bandwidths.

|  | 1 × Bandwidth | 2 × Bandwidth | 3 × Bandwidth |
|---|---|---|---|
| CEPI | 0.059 * | 0.064 * | 0.071 ** |
|  | (1.56) | (1.77) | (2.38) |
| Control variables | Yes | Yes | Yes |
| Time-fixed effect | Yes | Yes | Yes |
| Regional-fixed effects | Yes | Yes | Yes |
| N | 267 | 267 | 267 |

Note: *, and ** indicate $p < 0.1$, $p < 0.05$, respectively. T values are in parentheses.

## 5. Conclusions and Policy Recommendations

In recent years, the central environmental inspection policy has been an important means of promoting ecological civilization in China; it is an important means to correct the problem of "emphasizing legislation, neglecting law enforcement, and neglecting supervision" in the process of implementing environmental regulation via local governments and the predicament of environmental regulation failure caused by the "collusion of government and enterprises" by local governments, and it is of great practical significance to China's current environmental governance. However, there are few studies on the impact of the central environmental inspection policy on Chinese GTFP. Here, the regression discontinuity design method was used to examine the effect of the central environmental inspection policy on Chinese GTFP. The empirical results suggest that: (1) The Chinese central environmental inspection policy has a significant effect on the promotion of GTFP. (2) The mechanism of the central environmental inspection policy's impact on GTFP is brought about through significant TC; however, in the implementation of the central environmental inspection policy, the punishment of polluting enterprises, pollution rectification, and the promotion of the use of environmental technologies by manufacturing enterprises, conducting environmental inspections on manufacturing enterprises in various industries and the supervision of local governments by the environmental inspection teams will have a negative impact on technical efficiency. (3) The impact of the central environmental inspection policy on GTFP did not affect the main control variables, such as foreign investment, human capital level, industrial structure, and fiscal revenue. This shows that the central environmental inspection policy was the main reason for the increase in GTFP

during the year of its implementation. In addition, the increase in R&D capacity after the year of the policy's implementation may be due to an increased support for R&D by local governments under pressure from central environmental inspection policy authorities and the need to improve environmental performance after environmental inspections. (4) The effect of the central environmental inspection policy on TC is brought about by significantly increasing the magnitude of technological change. Although the policy has a positive effect on biased technological change, it is not significant. The policy's impact on TC is short-term to promote "pollution" in the production process; however, it has no significant effect on the allocation of resources, the expected output, and the proportion of pollution output, indicating that the technological nature of the production process has not been fundamentally reversed by the policy, that is, the jump-up effect of the Chinese central environmental inspection policy on GTFP is likely a short-term influence that does not significantly reverse the pollution output in the production process during the observation period, and is rather about regulating the act of polluting.

The above conclusions are of great theoretical and practical significance to the analysis of the effectiveness of the Chinese central environmental inspection policy and the subsequent implementation of environmental inspections. In response to the above conclusions, we give the following policy recommendations: (1) Give full play to the role of the central environmental inspection policy in correcting the "failure" of conventional environmental regulations, fully affirming the "authority" and effectiveness of the central environmental inspection policy central environmental inspection policy as an important means of correcting local governments' "emphasis on the economy and neglect of the environment". The role of the central environmental inspection policy in the coordination of government–enterprise relations in the process of local government governance will be brought into play to promote the environmental improvement of enterprise production and enhance the important functions of the technological progress. (2) The role of the central environmental inspection policy team should be limited to the scope of the planning of the policy for correcting the deviation of environmental regulations in the process of conducting inspections at the places where they are stationed, so as to avoid the expansion of the scope of environmental inspections and excessive administrative means, and to avoid the negative effects on management efficiency and production efficiency in the process of production to the greatest extent. (3) Improve the system of public participation in the implementation of the central environmental inspection policy, giving full play to the power of public supervision, and establishing a sound channel and complaint mechanism for the public to complain about environmental protection supervision, in order to enhance the interaction between the government and society in the process of local governments' environmental regulation, so that the central environmental inspection policy can become a continuous and regular environmental governance means, and so that the "short-term effect" of the central environmental inspection policy can be avoided. (4) Optimize the supervision mechanism of the central environmental inspection policy, integrate an ecological environment detection system in the process of central environmental inspection policy, establish a punishment system for pollution-related enterprises, and establish an incentive system for the green transformation of enterprises. To encourage enterprises' green technology research and development, we should not only strengthen the treatment technology of pollution emissions but also really strengthen the green transformation of enterprise production.

This study focuses on the meticulous analysis of the inherent mechanism of the impact that China's central environmental inspection policy exercises on green total factor productivity, based on Chinese urban data. Furthermore, it delves into the biased technological progress aspect and scrutinizes how this policy can promote the transition towards a greener economy. However, there are some defects in this study; for example, this study only analyzed the environmental protection policy for the promotion of the region, lacking manufacturers for micro-observation. Given that the implementation of the central environmental inspection policy targets production enterprises, the response measures these

companies take following environmental inspections serve as the most direct reflection of this environmental regulatory policy. The core goal of the central environmental inspection policy is to ensure that production enterprises upgrade their technology, update their equipment, and progress towards becoming more green. Therefore, the success of this policy primarily hinges upon the extent to which production enterprises can achieve this objective. In the future, we intend to examine the implementation effect of the central environmental inspection policy from the perspective of production enterprises.

**Author Contributions:** Conceptualization, M.D.; methodology, M.D.; software, M.D.; validation, M.D. and T.Z.; formal analysis, M.D.; investigation, D.W.; resources, M.D.; data curation, M.D.; writing—original draft preparation, M.D.; writing—review and editing, M.D.; visualization, M.D and T.Z.; supervision, D.W.; project administration, D.W.; funding acquisition, D.W. All authors have read and agreed to the published version of the manuscript.

**Funding:** This work was supported by the Social Science Planning Foundation of Liaoning Province under grant no. L22ZD065; the National Natural Science Foundation of China (Nos. 12271231, 72102129); and the Liaoning University Youth Research Fund Project under grant XJ2023004301.

**Institutional Review Board Statement:** Not applicable.

**Informed Consent Statement:** Not applicable.

**Data Availability Statement:** For the purpose of further research, this article does not provide data at this time.

**Acknowledgments:** The authors are grateful to the editor and the anonymous reviewers of this paper.

**Conflicts of Interest:** The authors declare no conflict of interest.

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
