# Peer review of "Can China’s Campaign-Style Environmental Regulation Improve the Green Total Factor Productivity?"

_sustainability, doi:10.3390/su152416902_

Round 1

Reviewer 1 Report

Comments and Suggestions for Authors

The peer-reviewed scientific study is undoubtedly a very interesting scientific work, the content of which can be beneficial not only for theory, but also for practice in the field of people management. At the outset, I can state that it has considerable potential, even if the content seems a bit chaotic to me and needs to be "done up". Likewise, the topic of shared management of people seems a little "timeless" to me, but you have to give it a chance.

The authors probably forgot to sufficiently read the instructions for authors available on the journal's website and strictly follow them so that the content structure is respected.

The abstract is not an introduction, but it should contain, according to the instructions for the authors, the objectives of the study and the material used - these two pieces of information are absolutely missing, the results of the study must be specified in the abstract.

Why aren't the keywords in alphabetical order and why are they capitalized?

The introduction lacks a clearly stated reason for writing this scientific study, a really clearly stated main goal and secondary goals. There is also a lack of stated hypotheses as well as research questions, which the authors would answer in the end and thus clearly fulfill the main meaning of the study.

Chapter 2 is mislabeled - contrary to instructions for authors.

The mandatory part of the discussion is missing, in the conclusion, as I already mentioned, there are no answers to the set research questions and hypotheses.

I also consider it necessary to explain the collection of data, to indicate in the tables whether it is your own data or taken over, or whether it is self-processing ....

To increase the scientific value of the work, I recommend the authors to expand the number of sources used (their number is absolutely insufficient for this type of scientific journal and scientific study) to include important ideas from current works by other than only Chinese authors

1.

Elena Mădălina PAŞCALĂU 2022. THE COMPETENCES OF PUBLIC ENVIRONMENTAL PROTECTION AUTHORITIES IN CASE OF NATURAL DISASTERS. Perspectives of Law and Public Administration. 11 (1). pp. 111-124

2. 

Peráček, T., Majerčáková, D. Mittelman, A. 2016. Significance of the waste act in the context of the right to protection of the environment. SGEM 2016 Conference Proceedings, Vol. 1, Book 5: Ecology and environmental protection. - Sofia : STEF92 Technology, 2016. - ISBN 978-619-7105-39-1. - S. 979-986 Paper presented at SGEM 2016 : International Multidisciplinary Scientific GeoConference SGEM 2016 [23rd] - Albena, 28.6.-6.7.2016. BG

  At least one paragraph about future research should also be added.

Reviewer 2 Report

Comments and Suggestions for Authors

The paper titled "Forestry Resources Efficiency, Can China's Campaign-Style Environmental Regulation Improve the Green Total Factor Productivity A Regression Discontinuity Design Based on the Central Environmental Inspection Policy " addresses an important and relevant topic - the empirically tests the internal influence mechanism of the central environmental protection inspection policy on green total factor productivity by using regression discontinuity design and tries to analyze its influence on technological progress from the perspective of biased technological progress. This article investigates an important research topic and contributes significantly to the existing literature. However, this study still needs a few improvements before its final acceptance.  Following are the suggestions for improvement.

Abstract: The abstract needs to arrange the goals, methods, and brief findings in the correct order. Make the necessary revisions. Ensure that the keyword sequence adheres to the guidelines set by the journal.

The introduction effectively sets the stage by providing a detailed context for the study. It explicitly outlines the objectives and methodology of the study, which involves assessing total factor productivity changes. This clarity helps readers understand the paper's focus. Simplify and condense the research problem, gaps, objectives, and innovative contribution to the study. Typically, authors conclude the introduction section by describing the study's innovations and its path. Revise to ensure the introduction section is appropriately structured. Follow the following paper: https://doi.org/10.1016/j.gr.2023.07.017

Literature Review:

Literature is advanced and relevant. However, if possible concise it and exclude irrelevant explanations. https://doi.org/10.1016/j.agwat.2023.108429

The research uses the DEA model to estimate productivity change in China. DEA is usually used when the data normality assumption is not met.  Authors need to add details about the suitability of DEA in this research instead of parametric techniques like SFA.

Carefully check the equation number.

Results and discussions are brief and explained in a systematic way.The conclusion and policy recommendations are too lengthy. Be specific to the concise and comprehensive study results and policy recommendations for Chinese provinces to improve their efficiency and productivity growth. Exclude unnecessary information in this section.

Avoid grammatical errors throughout the manuscript.

The caption of Table 4 does not seem appropriate. Change it.

Comments on the Quality of English Language

English is good enough

Reviewer 3 Report

Comments and Suggestions for Authors

Abstract: Title could be shortened even to just the first sentence. Some of the sentences are really long in the abstract and could be better clarified in shorter, more concise sentences, rather than run-on sentences throughout.

 Introduction: I didn’t see “Campaign Style” really defined until later in the article on Page 5.  All words expressed in the title and abstract should be discussed early on in the introduction. Line 28; there’s a juxtaposition; it should be “means of governance”. Page 2, line 61; end sentence with green economy, and start new sentence with Hao, et al., believe….Line 69 could end at spatial homogeneity and a new sentence start with “The promotion in…”Page 3: favour should be “favor”.

Background: Page 3, line 124; urgently should not be at the end of the sentence; try rewriting “to be urgently addressed”. Page 4, line 196; the findings by Wang are interesting; it’d be good to know the range of study years. Page 5 lines 218-222 too long sentence. 

Methods and Data: DEA (data envelope analysis?) and SBM both need to be defined as does a “relaxation variable”. Interestingly, DMU is correctly defined. I did a cursory examination of the equations in the following pages appear to be well done.   Equation (8) is a different font than the rest. Page 8 3.3 subtitle “Variables” should be capitalized. I’m accustomed to terms explained as “explanatory” variables. Page 9 line 352 should be local area. For lines 368 – 373 it would be better to move that paragraph to before the analysis section so readers may know where the data was sourced from, and how the variables were defined and measured, etc.

Empirical Analysis: Page 10 for Figures 1 and this comments pertains to all the figures; there need to be labels on both X & Y axes. Page 11, lines 435 and 436. ; should be a period. r & D should be R & D. Line 342 needs a period after “short term”. Throughout the paper, when I saw “short term” it would have really helped to understand how short term is defined and why long term wasn’t addressed?  Were the data years 2004-2018 representative of the “short term” so data wasn’t available for the long term?  Page 15 top lines should be moved under preceding page under Table 4. Also, for the sensitivity analysis, please provide the bandwidth estimates as footnotes to accompany the Table 5.

Conclusion: Good summarization of the results. For the policy recommendations, #4 makes good sense, however, there weren’t any of these policy recommendations analyzed in this paper, so that should be at least mentioned as a limitation of the research with opportunities for further research in the future.  

Overall comments: The writing is relatively clear but there needs to be another grammar check and consistencies in sentence structure, capitalizations of some words, etc. is needed.  Also, are there any limitations of this study? A short paragraph in the conclusions section can be added.

Comments on the Quality of English Language

Abstract: Title could be shortened even to just the first sentence. Some of the sentences are really long in the abstract and could be better clarified in shorter, more concise sentences, rather than run-on sentences throughout.

 Introduction: I didn’t see “Campaign Style” really defined until later in the article on Page 5.  All words expressed in the title and abstract should be discussed early on in the introduction. Line 28; there’s a juxtaposition; it should be “means of governance”. Page 2, line 61; end sentence with green economy, and start new sentence with Hao, et al., believe….Line 69 could end at spatial homogeneity and a new sentence start with “The promotion in…”Page 3: favour should be “favor”.

Background: Page 3, line 124; urgently should not be at the end of the sentence; try rewriting “to be urgently addressed”. Page 4, line 196; the findings by Wang are interesting; it’d be good to know the range of study years. Page 5 lines 218-222 too long sentence. 

Methods and Data: DEA (data envelope analysis?) and SBM both need to be defined as does a “relaxation variable”. Interestingly, DMU is correctly defined. I did a cursory examination of the equations in the following pages appear to be well done.   Equation (8) is a different font than the rest. Page 8 3.3 subtitle “Variables” should be capitalized. I’m accustomed to terms explained as “explanatory” variables. Page 9 line 352 should be local area. For lines 368 – 373 it would be better to move that paragraph to before the analysis section so readers may know where the data was sourced from, and how the variables were defined and measured, etc.

Empirical Analysis: Page 10 for Figures 1 and this comments pertains to all the figures; there need to be labels on both X & Y axes. Page 11, lines 435 and 436. ; should be a period. r & D should be R & D. Line 342 needs a period after “short term”. Throughout the paper, when I saw “short term” it would have really helped to understand how short term is defined and why long term wasn’t addressed?  Were the data years 2004-2018 representative of the “short term” so data wasn’t available for the long term?  Page 15 top lines should be moved under preceding page under Table 4. Also, for the sensitivity analysis, please provide the bandwidth estimates as footnotes to accompany the Table 5.

Conclusion: Good summarization of the results. For the policy recommendations, #4 makes good sense, however, there weren’t any of these policy recommendations analyzed in this paper, so that should be at least mentioned as a limitation of the research with opportunities for further research in the future.  

Overall comments: The writing is relatively clear but there needs to be another grammar check and consistencies in sentence structure, capitalizations of some words, etc. is needed.  Also, are there any limitations of this study? A short paragraph in the conclusions section can be added.

Reviewer 4 Report

Comments and Suggestions for Authors

This paper could be of great interest to the readers of this journal. However, some remarks may contribute to improve the quality of the paper:

1. Research design, questions, and hypotheses should be improved. For example, research questions and hypotheses should be stated from the beginning, and then now the results are achieved may contribute to answering the research questions, as well as testing the hypotheses. In this sense, research design can be improved.

2. Once the research design has improved, findings and the results discussion can be improved too.

3. Be sure that all references must be listed at the end of the paper.

Comments on the Quality of English Language

English is good. However, it is highly recommended to pass proofreading before considering for publication.

Reviewer 5 Report

Comments and Suggestions for Authors

Some suggestions for improvement:

1. Please shorten the title. Usually, a good title consists of approx. 14 words.

2. The main questions / hypothesis or statement of the problem this research addresses can be included and clarified within the abstract. 

3. Avoid too long sentences, e.g. lines from 156-162. Too long sentences may lose the content consistency. Check out the whole text in this regard.

4. The text within the lines from 156-194 is not referenced! Authors should be able to come to a shorter description of the policy situation and changes thereof. 

5. The formulas from the line 243 and beyond should be technically improved (Use formula editors and align the formulas with the text line)

6. The problem is well defined, and the Regression Discontinuity analyses are well displayed. However, there are no hypothesis, nor RQ formulated (expect in the title!) to address the problem. Clarify what exactly is the result of this analysis and relate this to the hypothesis resting. 

7. The authors have expressed the outcomes of the research in conclusions and recommendations; however, I would suggest making some consistency between: problem- hypothesis-results would make this research work more reliable and more understandable for the reader.

8. The literature reviewed is mostly from regional researchers. It is recommended to include some of world literature, since the problem discussed here is a worldwide problem. 

  •  

Round 2

Reviewer 1 Report

Comments and Suggestions for Authors

I am very glad that the authors fully accepted the comments of all reviewers and thereby contributed to the increased quality of their scientific study. I recommend an article for publication.

The proposed and accepted reference no. 28 indexed in the Web of Science and Scopus databases, however, must be completed so that it is linked to the cited work as follows:

Peráček, T., Majerčáková, D. Mittelman, A. 2016. Significance of the waste act in the context of the right to protection of the environment. ECOLOGY, ECONOMICS, EDUCATION AND LEGISLATION CONFERENCE PROCEEDINGS, SGEM 2016, VOL I, pp. 979-986, paper presented at 16th International Multidisciplinary Scientific Geoconference (SGEM 2016), JUN 30-JUL 06, 2016, Albena, Bulgaria
